# Endogenous Cardiac Steroids in Bipolar Disorder: State of the Art

**DOI:** 10.3390/ijms23031846

**Published:** 2022-02-06

**Authors:** Rif S. El-Mallakh, Vishnu Priya Sampath, Noa Horesh, David Lichtstein

**Affiliations:** 1Mood Disorders Research Program, Depression Center, Department of Psychiatry and Behavioral Sciences, University of Louisville School of Medicine, Louisville, KY 40202, USA; 2Department of Medical Neurobiology, Faculty of Medicine, The Institute for Medical Research, Israel-Canada, Hadassah Medical School, The Hebrew University, Jerusalem 9112102, Israel; vishnulifebt@yahoo.com (V.P.S.); noa.rosenthal1@mail.huji.ac.il (N.H.)

**Keywords:** bipolar disorder, endogenous cardiac steroids, endogenous ouabain, ouabain, Na^+^, K^+^-ATPase

## Abstract

Bipolar disorder (BD) is a severe psychiatric illness with a poor prognosis and problematic, suboptimal, treatments. Treatments, borne of an understanding of the pathoetiologic mechanisms, need to be developed in order to improve outcomes. Dysregulation of cationic homeostasis is the most reproducible aspect of BD pathophysiology. Correction of ionic balance is the universal mechanism of action of all mood stabilizing medications. Endogenous sodium pump modulators (collectively known as endogenous cardiac steroids, ECS) are steroids which are synthesized in and released from the adrenal gland and brain. These compounds, by activating or inhibiting Na^+^, K^+^-ATPase activity and activating intracellular signaling cascades, have numerous effects on cell survival, vascular tone homeostasis, inflammation, and neuronal activity. For the past twenty years we have addressed the hypothesis that the Na^+^, K^+^-ATPase-ECS system may be involved in the etiology of BD. This is a focused review that presents a comprehensive model pertaining to the role of ECS in the etiology of BD. We propose that alterations in ECS metabolism in the brain cause numerous biochemical changes that underlie brain dysfunction and mood symptoms. This is based on both animal models and translational human results. There are data that demonstrate that excess ECS induce abnormal mood and activity in animals, while a specific removal of ECS with antibodies normalizes mood. There are also data indicating that circulating levels of ECS are lower in manic individuals, and that patients with BD are unable to upregulate synthesis of ECS under conditions that increase their elaboration in non-psychiatric controls. There is strong evidence for the involvement of ion dysregulation and ECS function in bipolar illness. Additional research is required to fully characterize these abnormalities and define future clinical directions.

## 1. Introduction

Type I bipolar disorder (BD) is a severe psychiatric illness that manifests as extreme variations in mood and energy, usually labelled as mania and depression, interspersed over a euthymic or dysthymic baseline [1]. The disorder afflicts approximately 1% of people [2,3], with documented suboptimal treatments and a host of undesirable outcomes related to both the disease and its treatment [4,5]. Despite over 60 years of directed effort, the pathoetiology of the illness remains unknown [3], but multiple clues have emerged that continue to inform ongoing research. The illness is viewed as multifactorial, involving elements (of development and neuroplasticity, inflammation, and aberrant modulation of brain function and circuitry) that are mediated by gene and environment interaction through multiple inherited genes and multiple altered epigenetic changes [3,6,7,8]. However, the absence of a centralized unifying model contributes to the fragmented, siloed, fashion of current pathophysiologic research. In this article, we propose a mechanism of pathophysiology that incorporates much of the translational and clinical data, and, in addition to focusing research efforts, also leads to potential novel treatment options.

Among the most reproducible findings in bipolar illness, we observe the dysregulation of control of the electrically important ions sodium (Na^+^), potassium (K^+^), hydrogen (proton, H^+^), and calcium (Ca^2+^) [9]. Ion regulation spans across all of the proposed mechanisms pertaining to pathogenesis of abnormal moods in bipolar illness, and it spans across all successful treatment options. For example, in an analysis of susceptibility loci, Askland found that approximately 74% of known loci involve genes related to ion regulation, or what she refers to as neuroelectrical genes [10]. By comparison, genes involved in any neurotransmitter pathway account for only about 58% of the susceptibility loci, and the monoamines account for only 31% [10]. A similar conclusion is reached when genome wide association studies (GWAS) are explored [11]. The only animal models of bipolar illness, meeting all validity criteria, create ion transport abnormalities [12,13,14]. Additionally, nearly all interventions, effective in mania or mood stabilization, reduce intracellular sodium either directly or indirectly [15,16,17]. Changes in sodium pump activity are associated with changes in mood state [18,19], and endogenous cardiac steroids (ECS), that can determine sodium pump activity, are dysregulated in patients with BD [11,20].

In this review, we summarize the data implicating ECS in the pathophysiology of BD, determine necessary future directions, and reflect on treatment development as a consequence of this work.

## 2. Results and Discussion

Cardenolides, such as ouabain and digoxin, and bufadienolides, such as bufalin, are steroids originally identified in plants (*Digitalis, Strophantus*) and toads (*Bufo*), which have been used for hundreds of years in Western and Eastern medicine to treat heart failure, arrhythmias, and other maladies. In the past 25 years, the cardenolides ouabain and digoxin, and the bufadienolides 19-norbufalin, marinobufagenin, and cinobufagenin were identified as normal constituents of mammalian tissues, including the brain [21,22,23,24]. These compounds, collectively termed endogenous cardiac steroids (ECS), are synthesized in the adrenal and hypothalamus of mammals [25,26,27], and they are considered a new class of hormones implicated in many physiological and pathophysiological mechanisms, including cell growth and cancer, vascular tone homeostasis, blood pressure, hypertension, natriuresis, heart contractility, and inflammation [22,24,28].

These mammalian ECS resemble non-mammalian compounds in their ionotropic effects, presumably due to inhibition of the sodium, and potassium-activated adenosine triphosphatase, or sodium pump (Na^+^, K^+^-ATPase) [29]. However, at low physiologic concentrations (nM), these compounds have been shown to stimulate pump activity [30,31,32,33]. Importantly, the interaction of ECS with the Na^+^, K^+^-ATPase results not only in the inhibition of the ion pumping function, but causes the activation of several signal transduction cascades, including mitogen-activated protein kinase (MAPK), extracellular signal-regulated kinase (ERK), protein kinase B, and oroto-oncogene tyrosine-protein kinase pathways [34,35]. In addition, a substantial body of studies has demonstrated that ECS also act by directly and indirectly affecting the activity of the nuclear receptor superfamily of transcription factors [36,37].

To date, the most studied ECS is the endogenous ouabain (EO); a compound that is similar or identical to the plant steroid. Based on its immunoreactivity with anti-ouabain antibodies, this compound has been shown to be present in mammalian brain and CSF, and it is considered a potential neuromodulator [22,38,39,40].

It is important to note that there remains some debate regarding the exact structure of EO [41]. Central to this debate, there is difficulty measuring the picomolar concentrations that exist in mammalian systems using direct chemical methods [42]. Notably, the evidence for the existence of EO is quite overwhelming, and the dispute is focused on the exact structure of this steroid [27].

The synthetic pathways of ECS are not established, but it is known that cholesterol (which may be reduced in the brains of patients with bipolar illness [43]) is needed for ECS/EO production [27], and the pathway may involve pregnenolone and progesterone as intermediate steps [42]. Pregnenolone levels may be reduced in the cerebrospinal fluid of individuals with a diagnosis of mood disorder, and this reduction may be in relation to the severity of their symptoms [44]. Rapid elevations in plasma levels of ECS, as occurs with exhaustive exercise of normal controls [45], may occur due to the release of bound steroids from a carrier protein [46].

Despite the limited available data on the metabolism of ECS in general, and EO in particular, these compounds are referred to by some as “a hormone family” [21,27,47,48,49,50,51,52]. However, at the same time, as is evident from the lack of reference in textbooks and reviews, these steroids are ignored almost completely by mainstream biochemists, physiologists, and endocrinologists. We pointed out the crucial importance of deciphering the biosynthetic pathway of ECS close to ten years ago [53], but this issue remained the Achilles heel of this field of research.

At *experimental* or *pharmacologic* micromolar concentrations, exogenously administered ouabain inhibits the activity of the sodium pump. However, at *physiologic* picomolar to nanomolar concentrations, EO *increases* sodium pump activity [30,31,32,33]. In notable cases, when there are documented increases of the plasma levels of EO, the total levels remain below concentrations that inhibit sodium pump activity [54]. At the low concentrations, EO also appear to activate second messengers of the Src kinase-, ERK1/2-, and Akt-mediated pathways or the sodium-proton exchanger 1 (NHE1) [54,55,56,57].

Activation of the sodium pump appears to be an essential feature in reducing central nervous system inflammation [58]. If sodium pump activity is blocked in glial cells, inflammatory pathways are activated in the presence of lipopolysaccharides [58].

### 2.1. Role of Exogenous CS and ECS in Animal Models of Mood Disorders

The process of modeling psychiatric disorders in animals is difficult because of the subjective nature of many psychiatric symptoms. Robbins and Sahakian set forth three requirements for an acceptable animal model of a psychiatric disorder [59]. First, they proposed “face validity,” in which they noted that the model must share the pathophysiological changes known to occur in the human condition. Second, they required “construct validity,” for which they required that the model display similar behavioral manifestations as the human disease. Finally, they defined “predictive validity,” in which the induced abnormal behavior responds to medications that improve the symptoms in afflicted humans [59].

The most commonly used model of mania pertains to administration of amphetamine to rats [60] or mice [61]. While this model does not fulfill Robbins and Sahakian’s criteria (it is actually an animal model of substance abuse), it has long been used as *the* animal model for mania. Mice receiving 5 mg/kg of amphetamine, by intraperitoneal (IP) injection, developed hyperactivity immediately after injection; however, co-administration of intracerebroventricular (ICV) anti-ouabain antibodies normalized the amphetamine-induced hyperactivity [62]. ICV anti-ouabain antibodies alone had no effect [62]. Amphetamine was associated with a 300% increase in brain levels of ECS [62]. This may increase Na^+^, K^+^-ATPase activity in the brains of rats [63], or reduce Na^+^, K^+^-ATPase activity if the rats were fed a diet, rich in hydrogenated vegetable fats, which is rich in *trans* fatty acids [64]. Additionally, amphetamine, when administered to rhesus monkeys, upregulates the α3 isoform of the sodium pump [65]. In mice, ICV administration of anti-ouabain antibodies appears to inhibit the increase in brain ECS while preventing the associated hyperactivity [62].

Similarly, sterile inflammation appears to be a characteristic of bipolar disorder pathophysiology [66]. Chronic inflammation may induce oxidative stress [67]. In addition to playing a role in the genesis of symptoms, these conditions may also be responsible for comorbid medical conditions [66,68]. Cardiac steroids appear to mediate elaboration of oxidative stress [34], and the dramatic increase of EC in amphetamine treated rats may increase inflammation and oxidative stress as reflected by the increase of superoxide dismutase (SOD) activity in these animals [69]. Treatment with anti-ouabain antibodies normalizes SOD in amphetamine-treated mice [69].

Despite these findings pertaining to the amphetamine model, animal models in which the sodium pump is rendered dysfunctional, either by using ICV injections of ouabain [70,71,72], or by knocking down the α [73] or α3 [74,75,76] isoforms of the sodium pump, fulfill all three validity criteria for an animal model of a psychiatric illness [13]. These models mimic the abnormalities documented in bipolar patients, including reduced sodium pump activity, increased intracellular sodium [13], and CNS hypometabolism [77]. Importantly, these are the only animal models in which the same abnormality can produce both manic-like behaviors [71] and depressive-type behaviors [78]; thus, they are the best models we have of BD [13].

In Flinders Sensitive Line (FSL) of genetically depressed rats, ICV administration of 50 μg of anti-ouabain antibodies significantly reduced depressive behaviors [79]. These same antibodies halved the concentration of ECS in ex vivo samples of human cerebrospinal fluid (SCF) [79].

A reduction of sodium pump activity, via administration of exogenous ouabain or by genetic knockout, serves as an animal model of mania and depression. Conversely, removal of ECS in animals with anti-ouabain antibodies appears to reduce manic symptoms. While these two observations appear to contradict each other, it should be noted that the former (reducing sodium pump) is being performed on a normal animal, while the later (removing EO with anti-ouabain antibodies) is being performed on a symptomatic animal.

### 2.2. Role of ECS in Humans Mood Disorders

ECS appear to play some role in the pathophysiology of BD. Patients with type I bipolar illness display reduced circulating EC when they are manic compared to healthy controls (143.6 ± S.E.M. 20.94 vs. 296.6 ± 12.76 pg digoxin equivalents/mL, respectively, F = 4.77, *p* < 0.05), but not compared to euthymic bipolar subjects (213.8 ± 86.92, *p* = 0.77) [80]. Furthermore, bipolar patients appear to be unable to increase production of EO based on physiologic need. They do not display the seasonal variation in EO levels that has been demonstrated in healthy controls (higher concentrations in the Spring, Summer, and Fall, and lower levels in Winter, versus low levels all year long in patients with bipolar disorder [80]). More importantly, euthymic patients with bipolar illness do not increase circulating EO in the setting of exercise to exhaustion, as occurs with matched non-bipolar controls [81] or the general population [45]. It has been proposed that sleep deprivation, which is associated with increased production of EO and corticosterone in mice [82], might be a trigger for a manic switch in bipolar subjects [21,83]. As little as 24 h of sleep deprivation in humans is enough to increase serum cortisol [84,85,86] and manic-like symptoms [87].

There is direct evidence pertaining to the involvement of both the sodium pump and ECS in the CNS of patients with BD. An allelic association between BD and a Na^+^, K^+^-ATPase α3 subunit gene (*ATP1A3*) has been reported [88]. In comparison to non-psychiatric controls, the α3 subunit may be specifically under-expressed in neurons that release gamma aminobutyric acid (GABA) in the parietal cortex of patients with bipolar illness [89]. The parietal cortex of patients with bipolar illness may have fewer ouabain binding sites than non-psychiatric controls [90]. Subsequently, we have demonstrated a significant association with BD of six single SNPs in the three genes of the Na^+^, K^+^-ATPase α isoforms. Haplotype analysis of the α2 isoform showed a significant association with two loci haplotypes pertaining to BD, suggesting that this enzyme plays a role in the etiology of the disease [91], and, indeed, this isoform appears to be reduced in the temporal cortex of individuals with bipolar disorder [92], but α2 and α3 may be increased in the prefrontal cortex [20].

When EO has been measured directly in the prefrontal cortex (PFC), it has been found to be at the same level as non-psychiatric controls; however, this finding appears to be driven by two outlier patients with EO levels more than three times higher than the average for the entire group [20]. Interestingly, smoking is associated with reduced PFC EO levels [20], and this is important because individuals with bipolar illness are more than twice as likely to smoke as the general population [93]; the smoking status of the two outliers is not known [20].

### 2.3. Hypothesis

Alterations of EO levels in bipolar disorder appear to be important in the pathogenesis of the illness. There is evidence for both an increase or a decrease in EO concentrations, and there are reasons to believe that each change can lead to the symptoms of the illness (Figure 1). This is an appropriate time to begin the process of translation from basic to clinical research.

### 2.4. Translational Implications: Pathophysiologic Models

The data suggest two parallel but, possibly, competing models. In the first model, some factor (inflammation or physiologic stress) results in increased production of central nervous system EO.

As mentioned above, ECS are known to activate the Src, ERK, and AKT signaling pathways [22,94,95,96]. These are all signaling systems that have been implicated in BD [97,98,99,100]. Additionally, inhibition of sodium pump activity [70] and altering sodium pump isoform expression [72] have been demonstrated in animals receiving ICV ouabain. Activation of signaling pathways occur at levels that do not inhibit sodium pump activity, and this may be important in inducing symptoms of BD. Inactivation of EO with anti-ouabain antibodies would be a reasonable treatment option in this model of relative EO excess (Figure 1).

The second model may have a very similar physiology. In this model, some factor (inflammation or physiologic stress) results in a signal to increase central nervous system EO. However, patients with BD disorder are incapable of upregulating EO under conditions of high demand, and the body experiences a relative EO deficiency (Figure 1). Exogenous ouabain administration would be a reasonable treatment option in this model. Each one of these models will be discussed in additional detail.

### 2.5. EO Excess

Direct measurements of EO in the parietal cortex of postmortem samples have shown higher levels in BD patients, as compared to those with schizophrenia, major depression, and psychiatrically normal individuals [90]. The increased levels of EO in the brain of BD patients are likely to be brain-region specific, as this increase has not been witnessed in the temporal cortex [89]. Results from one of our laboratories demonstrate that the reduction in brain EO (by intracerebroventricular administration of anti-ouabain antibodies) creates a dramatically protective effect pertaining to a depressive-like behavior in rats [90], with concomitant alterations in catecholamine levels in specific brain regions [79]. Furthermore, a reduction in brain EO also protects against a manic-like behavioral response in an amphetamine (AMPH)-induced mania model [62], with a concomitant reduction of oxidative stress markers [69].

In view of the above, we predicted that the reduction of circulating EO in BD patients might ameliorate their symptoms. Indeed, in a preliminary study, Digibind (Digoxin Immune Fab fragment used for the treatment of digoxin intoxication) was given to six inpatients who were hospitalized for a recurrent episode of bipolar depression. The clinical rating of depression was ascertained using the Montgomery—Asberg Depression Rating Scale (MADRS), before the intervention, and 6, 24, 48, and 72 h after the intervention. In the results of this experiment, there was an observed decrease in the MADRS score, which reached a peak (8.3 points) 24 h after the administration of Digibind, followed by a gradual return towards baseline scores over the following two days (Zilberstein, A., Krivoy, N., Klein, E. and Lichtstein, D. unpublished observation). These preliminary results, which need to be characterized and extended to more patients, strongly suggest that increased levels of EO are involved in the pathophysiology of the disease and, consequently, their reduction will have beneficial effect.

The mechanisms underlying BD are extremely complex, and the increased ECS levels may be involved in this pathology at various stages and in different biochemical pathways. As mentioned above, it is well established that ECS cause both the inhibition of Na^+^, K^+^-ATPase transport activity and the activation of the Ras-Raf-MEK-ERK signaling cascade, resulting in the activation of MAPK [34]. These two events lead to an increase in intracellular Ca^2+^ [101], resulting in the opening of mitochondrial ATP-sensitive K^+^ channels and the generation of reactive oxygen species (ROS) [102]. The increase in ROS levels, beyond the capacity of antioxidant defense mechanisms in the brain, damages cellular components, leading to a disruption of normal neuronal function, and this may contribute to the pathological behavioral changes which are characteristic of BD.

### 2.6. Relative EO Deficiency

As previously noted, work in one of our laboratories has identified reduced plasma levels of EO in unmedicated manic patients versus unmedicated controls [80]. Additionally, individuals with BD lack the seasonal variation of EO levels that has been noted in the control subjects who display higher levels in the spring, summer, and autumn (in the range of 400–600 pg digoxin equivalents/mL), and lower levels in the winter (around 150 pg digoxin equivalents/mL), with levels in the range of 150 pg digoxin equivalents/mL throughout the entire year [80]. More importantly, euthymic patients with BD do not increase EO levels when they exercise to exhaustion, as has been noted to occur in non-bipolar controls [45] (EO at 60 min of exercise 0.007 ± S.D. 0.02 ng/mL in bipolar vs. 0.075 ± 0.06 ng/mL in normal control subjects, *p* = 0.029, but not significant at peak exercise, 0.009 ± S.D. 0.02 ng/mL in bipolar vs. 0.131 ± 0.21 ng/mL in normal control subjects, *p* = 0.15 [81]). In these studies, BD patients have measurable EO, but it does not vary with physiologic need.

It is proposed that physiological elevations in EO levels that may normally occur with inflammation, severe illness [103], or sleep deprivation [82], do not occur in subjects with BD, and they experience a relative EO deficiency. Considering that physiologic levels of EO activate Na^+^, K^+^-ATPase activity, the relative deficiency results in sodium pump hypofunction in relation to the need, and consequent depolarization of neural tissues [18,21].

### 2.7. Future Work

The process of utilizing the EO system as a potential treatment for bipolar disorder holds a lot of promise, but several studies must be performed prior to the proposal of definitive human studies. They include the following:Replication and confirmation of peripheral EO levels, in a larger population of patients with BD both while they are ill and while they are euthymic, are needed to ensure the generalizability of the initial studies. Additionally, the specificity of this effect must be confirmed by examining both psychiatrically ill controls (major depression and schizophrenia), and psychiatrically healthy controls.Examination of brain area-specific EO concentrations in patients and controls. This would utilize postmortem tissues, and this overall process is necessary to make sure that peripheral measurements are reflected in the central nervous system. This is needed to determine if there are brain region-specific alterations. The envisioned treatment protocols are not region-specific, therefore, anatomical localization will help understand untoward side effects of treatment.Utilize neuronal cells obtained from patients and controls (e.g., induced pluripotent cells [104] or olfactory neuroepithelial cells [105]) in order to determine the differential effect (if any) in the response of neural tissue derived from patients with BD and non-psychiatric controls. Early exploratory evidence from immortalized lymphoblasts [106], induced pluripotent cells, and olfactory neuroepithelial progenitors [107,108,109,110,111], suggests that such differences exist.

As soon as these questions have been answered, clinical studies, with anti-ouabain antibodies (for relative EO excess) or exogenously administered ouabain (for relative EO deficiency; dosing and pharmacokinetics to be determined), can be planned.

## 3. Methods

This is a focused narrative review [112] that presents a comprehensive model of the role of ECS in the etiology of BD. A literature search was conducted in the PubMed database, for clinical and basic research studies, and review articles, published in the time period occurring until December 2021. The PubMed database was searched using the MeSH terms, alone or in combination: ‘bipolar disorder’, ‘bipolar disease’, ‘ouabain’, ‘endogenous ouabain’, ‘cardiac steroids’, and ‘endogenous cardiac steroids’. In addition, reference lists of identified papers were manually checked for additional related articles. Due to f the comprehensive nature of this review, it includes important historical work, and it draws significantly from the authors’ own work because they have been involved in this work for decades.

## 4. Conclusions

Alterations in ECS metabolism in the brain causes numerous biochemical changes that underlie brain dysfunction and mood symptoms. This finding is based on both animal models and translational human results. There are data that demonstrate that excess ECS induce abnormal mood and activity in animals, and there are data suggesting that specific removal of ECS with antibodies normalizes mood. There are also data indicating that circulating levels of ECS are lower in the plasma of manic individuals, and there are data indicating that patients with BD are unable to upregulate synthesis of ECS under conditions that increase their elaboration in non-psychiatric controls. There is strong evidence for the involvement of ion dysregulation and ECS function in bipolar illness. Additional research is required in order to fully characterize these abnormalities and define future clinical directions.

## Figures and Tables

**Figure 1 ijms-23-01846-f001:**
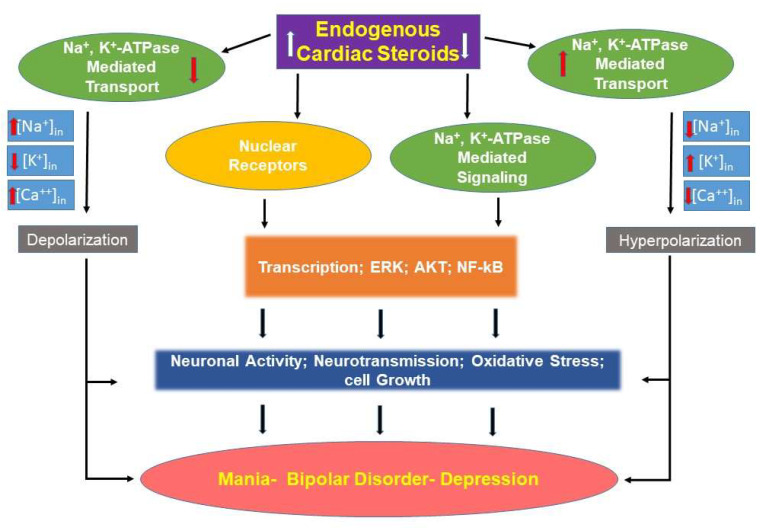
Schematic representation of the possible links between endogenous cardiac steroids and bipolar disorder. See text for details.

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
