# Peer review of "Endogenous Cardiac Steroids in Bipolar Disorder: State of the Art"

_ijms, 2022, doi:10.3390/ijms23031846_

Round 1
Reviewer 1 Report
Dear Authors,
I have read with great interest your review titled: Endogenous cardiac steroids in bipolar disorder: State of the art".
However, based on the content, compared with the existing literature, this review is naive and should have a clear scope from the very beginning.
Minor points:
The entire manuscript should be checked once again for typos.
Carefull on the citations. They are doubled.
Please divide each long paragraph (>10 rows) into small ones (5-6 rows). It can be tiring for the reader.
Major points:
What new findings, novel perspective brings this manuscript compared to those existing in the literature?
The authors stated that this is a focused review also based on the author's own work. But this does not mean is it auto plagiarism? Why not a narrative review as described by Green et al., ("Writing narrative literature reviews for peer-reviewed journals: secrets of the trade").
There is no searching strategy, keywords used for studies identification, databases used to identify suitable articles, inclusion/exclusion criteria, and limitations of the study.
Excepting 39 references that were published in the last 5/6 years, the rest are older which can impact the validity and overall quality of your manuscript. Also, I would have like to see a table stratified based on the cell line/experimental model used. Kind regards and all the best, The ReviewerAuthor Response
We thank the reviewers for their important and constructive comments. All the concerns raised by the reviewers were addressed and corrected.
Reviewer 1:
Minor points:
- The entire manuscript should be checked once again for typos.
Response: We reviewed the manuscript again. Please note that we have used American spelling. I suspect the reviewer may have been expecting British spelling.
- Careful on the citations. They are doubled.
Response: The citations were corrected.
- Please divide each long paragraph (>10 rows) into small ones (5-6 rows). It can be tiring for the reader.
Response: The long paragraphs were divided into smaller sections as much as possible.We could not reduce all paragraphs to only 5-6 rows, but we did break up overly long paragraphs in lines 81, 109, 203, and 284.
Major points:
- What new findings, novel perspective brings this manuscript compared to those existing in the literature.
Response: This is, to the best of our knowledge, the first attempt to review all the available literature on the possible involvement of endogenous cardiac steroids in bipolar disorder. This issue was indeed addressed partially in previous publications of the authors but a comprehensive review including unique schematic representation was not performed.
- The authors stated that this is a focused review also based on the author's own work. But this does not mean is it auto plagiarism? Why not a narrative review as described by Green et al., ("Writing narrative literature reviews for peer-reviewed journals: secrets of the trade").
Response: Indeed, this is a Narrative overview and is in accord with the proposed presentation by Green et al. The drive of this review is to “pull many pieces of information together into a readable format. They are helpful in presenting a broad perspective on a topic and often describe the history or development of a problem or its management…”. We have added the Green reference to highlight the methods.
- There is no searching strategy, keywords used for studies identification, databases used to identify suitable articles, inclusion/exclusion criteria, and limitations of the study.
Response: We clarified our methods and modified the Methods section to reflect this: “This is a focused narrative review [111] that presents a comprehensive model of the role of ECS in the etiology of BD. A literature search was conducted in the PubMed database for clinical and basic research studies and review articles up to December 2021. The database PubMed was searched using the MeSH terms, alone or in combination: ‘bipolar disorder’, ‘bipolar disease’, ‘ouabain’, ‘endogenous ouabain’, ‘cardiac steroids’, ‘endogenous cardiac steroids’. In addition, reference lists of identified papers were manually checked for additional related articles.”
- Excepting 39 references that were published in the last 5/6 years, the rest are older which can impact the validity and overall quality of your manuscript.
Response: Since, the studies reviewed were published in the past 15 years, it is not surprising that many of the references were published in more than 6 years ago. We added clarification in the Methods section: “Because of the comprehensive nature of the review, it includes important historical work and draws significantly from the authors’ own work because they have been involved in this work for decades.”
- Also, I would have like to see a table stratified based on the cell line/experimental model used.
Response: We created the table below for the reviewer, but we do not believe that such a table would actually be appropriate for the paper. We believe that the focus of the paper is endogenous cardenolides, and while be believe that cellular models are important, that is not the focus of the paper.
Table for reviewer to look at. |
|||
Cell Model |
Attributes |
Advantages |
Disadvantages |
Induced pluripotent cell lines |
1. Minimally invasive biopsy 2. Possess genetic heritage of disorder 3. Essentially permanent 4. Requires genetic dedifferentiation 5. Requires chemical/genetic differentiation 6. Labor intensive 7. Documented utility |
1. More widely available and understood by more researchers.
|
1. Must be dedifferentiated, which involves induced changes in many genes and reduces generalizability |
Olfactory neuroepithelial cells lines |
1. Minimally invasive biopsy 2. Possess genetic heritage of disorder 3. Essentially permanent 4. Does not require genetic dedifferentiation 5. Requires chemical/genetic differentiation 6. Labor intensive 7. Documented utility |
1. Dedicated neuroprogenitor cells that will spontaneously differentiate into neurons and glia 2. Will spontaneously produce a mixed culture that more closely resembles the brain |
Used by only a few laboratories. |
Reviewer 2 Report
The manuscript entitled: “Endogenous cardiac steroids in bipolar disorder: State of the art” by Rif S El-Mallakh et al. (Manuscript ID: ijms 1541186) is a review of the current knowledge of the role of the sodium pump (Na+, K+ -ATPase)/endogenous cardiac steroids (ECS) system in the pathophysiology of bipolar disorder (BD).
This is an interesting and well written review with interesting hypothesis. I have no question for the authors.
Author Response
No changes were made in response to this review. We appreciate the reviewer taking the time and effort to review our submission.
Round 2
Reviewer 1 Report
Dear Authors,
Congratulations on the work on reviewing your manuscript. Indeed, you answer all my raised issues, and Figure 1 significantly helps the reader to understand the mechanisms. Before being accepted, please remove "Table 1" from the manuscript, line 310 if you suggested that Table 1 from the Response Letter is only indicative. However, I still feel that a Table is needed.
Kind regards,
The Reviewer
Author Response
We appreciate the reviewer and their input. We eliminated the reference to Table 1 in the text. Thank you.
